# Dietary Polyunsaturated Fatty Acid Deficiency Impairs Renal Lipid Metabolism and Adaptive Response to Proteinuria in Murine Renal Tubules

**DOI:** 10.3390/nu17060961

**Published:** 2025-03-10

**Authors:** Yaping Wang, Pan Diao, Daiki Aomura, Takayuki Nimura, Makoto Harada, Fangping Jia, Takero Nakajima, Naoki Tanaka, Yuji Kamijo

**Affiliations:** 1Department of Metabolic Regulation, Shinshu University School of Medicine, Matsumoto 390-8621, Japan; 19101672@hebmu.edu.cn (Y.W.); jiafangping2022@163.com (F.J.); 21hm122a@shinshu-u.ac.jp (T.N.); naopi@shinshu-u.ac.jp (N.T.); 2Basic Nursing, Hebei Medical University, Shijiazhuang 050017, China; 3Department of Clinical Laboratory, The Second Hospital of Hebei Medical University, Shijiazhuang 050017, China; 29201095@hebmu.edu.cn; 4Postdoctoral Mobile Station of Clinical Medicine, Hebei Medical University, Shijiazhuang 050017, China; 5Department of Nephrology, Shinshu University School of Medicine, Matsumoto 390-8621, Japan; aomura91@gmail.com (D.A.); nakat@shinshu-u.ac.jp (T.N.); tokomadaraha724@gmail.com (M.H.); 6Department of Infectious Diseases, The First Affiliated Hospital, Zhejiang University School of Medicine, Hangzhou 310003, China; 7Center for Medical Education and Clinical Training, Shinshu University School of Medicine, Matsumoto 390-8621, Japan; 8Department of Global Medical Research Promotion, Shinshu University Graduate School of Medicine, Matsumoto 390-8621, Japan; 9International Relations Office, Shinshu University School of Medicine, Matsumoto 390-8621, Japan; 10Research Center for Social Systems, Shinshu University, Matsumoto 390-8621, Japan

**Keywords:** PPARα, fatty acid metabolism, lysosome, tubular protein reabsorption, kidney function

## Abstract

**Background/Objectives**: Kidneys are fatty acid (FA)-consuming organs that use adenosine triphosphate (ATP) for tubular functions, including endocytosis for protein reabsorption to prevent urinary protein loss. Peroxisome proliferator-activated receptor α (PPARα) is a master regulator of FA metabolism and energy production, with high renal expression. Although polyunsaturated fatty acids (PUFAs) are essential nutrients that are natural PPARα ligands, their role in tubular protein reabsorption remains unclear. As clinical PUFA deficiency occurs in humans under various conditions, we used a mouse model that mimics these conditions. **Methods**: We administered a 2-week intraperitoneal protein-overload (PO) treatment to mice that had been continuously fed a PUFA-deficient diet. We compared the phenotypic changes with those in mice fed a standard diet and those in mice fed a PUFA-deficient diet with PUFA supplementation. **Results**: In the absence of PO, the PUFA-deficient diet induced increased lysosomal autophagy activation; however, other phenotypic differences were not detected among the diet groups. In the PO experimental condition, the PUFA-deficient diet increased daily urinary protein excretion and tubular lysosomes; suppressed adaptive endocytosis activation, which was probably enhanced by continuous autophagy activation; and worsened FA metabolism and PPARα-mediated responses to PO, which disrupted renal energy homeostasis. However, these changes were attenuated by PUFA supplementation at the physiological intake level. **Conclusions**: PUFAs are essential nutrients for the tubular adaptive reabsorption response against urinary protein loss. Therefore, active PUFA intake may be important for patients with kidney disease-associated proteinuria, especially those with various PUFA deficiency-inducing conditions.

## 1. Introduction

Proximal tubular epithelial cells (PTECs) are highly differentiated cells that absorb urinary molecules. Among the diverse reabsorptive mechanisms of various urinary substances, the endocytosis–lysosomal degradation system is important for the reabsorption of urinary proteins, although it requires a large amount of ATP for its smooth functioning. The kidney is a representative fatty acid (FA)-consuming organ. PTECs contain numerous mitochondria and mainly use FA as an energy substrate to produce large amounts of ATP and maintain their reabsorptive function.

The peroxisome proliferator-activated receptor α (PPARα), a member of the nuclear receptor superfamily, is a master regulator of FA metabolism and mitochondrial energy production. PPARα activation increases the expression of various FA metabolism-related factors, such as mitochondrial β-oxidation enzymes [1,2,3]. Moreover, PPARα participates in various pathophysiological processes, including cell proliferation [4], glucose metabolism [5,6], inflammation [7,8], oxidative stress [9], apoptosis [10], and autophagy [11,12]. PPARα is highly expressed in the kidney and mainly localized in the proximal tubules [13,14,15]. Studies in starved *Ppara^−/−^* mice demonstrated an important physiological role of PPARα in maintaining FA metabolism and ATP homeostasis in the tubules and supporting tubular protein reabsorptive function through the endocytosis–lysosomal degradation flux [16].

Polyunsaturated fatty acids (PUFAs) constitute a class of FAs that contain two or more unsaturated carbon double bonds, which predominantly include linoleic acid (18:2, *n*-6) and α-linolenic acid (18:3, *n*-3). Given their lack of desaturases, mammals cannot synthesize these essential FAs and, therefore, must obtain them from food or supplements. PUFAs are the natural ligands for PPARα, and thus, PUFA deficiency decreases the adaptive responsiveness of PPARα [17]. However, it is unclear whether a PUFA-deficient diet reduces PPARα function in the kidney and affects the tubular protein-reabsorption process. The primary objective of the study was to determine whether PUFA deficiency affects the adaptive tubular response against proteinuria and to elucidate the underlying mechanisms. The secondary objective was to clarify the importance of adequate PUFA intake in the adaptive tubular response against proteinuria, after excluding confounding variables. PUFA deficiency occurs in humans under various conditions, with kidney disease being a common manifestation. To ascertain the influence of PUFA deficiency in humans, we used a mouse model that mimics these conditions.

## 2. Materials and Methods

### 2.1. Methods for Clarifying Research Questions

To clarify the primary objective, we intraperitoneally administered protein to induce protein overload (PO) in mice that had been continuously fed a PUFA-deficient diet and then compared the phenotypic changes in these mice with those in mice that had been fed a standard diet. The PO treatment is an experimental system wherein excess intraperitoneal external protein is absorbed into the circulatory system and excreted from the glomeruli during primary urinary filtration, with consequent excessive PO in the renal tubules. This experimental system can be used to investigate the adaptive tubular protein-reabsorptive responses to excess proteinuria.

To clarify the secondary objective, we examined whether PUFA supplementation at the physiological intake level improves the phenotypic characteristics of PO-treated mice that were fed a PUFA-deficient diet. To standardize the calorie intake and total fat consumption in the PUFA-deficient and PUFA-supplemented groups, each group of mice received the same caloric content of PUFA-deficient or PUFA-containing oil in addition to the same PUFA-deficient diet.

### 2.2. Diets, Animals, and Treatments

The animal experiments adhered to the animal study protocols approved by the Shinshu University School of Medicine (Approval Number: 290064/1 December 2017) before the study and were performed in accordance with the National Institutes of Health (NIH) Guide for the Care and Use of Laboratory Animals. The present study was not registered in an animal study registry because it was not an essential requirement for study conduct. For this study, 8- to 9-week-old C57BL/6 strain mice were obtained from Charles River Laboratories (Yokohama, Japan). The mice were maintained in a controlled environment (22–23 °C; 12 h light/dark cycle) with a 5% weight/weight (*w*/*w*) crude fat-containing standard rodent chow (MF; Oriental Yeast, Tokyo, Japan) and water during the 1-week acclimatization period. The mice were then randomly divided into the control (standard diet, *n* = 4), PUFA (+) diet (PUFA supplementation on a PUFA-deficient diet, *n* = 4), and PUFA (−) diet (continuously PUFA-deficient diet, *n* = 4) groups to investigate the renal effects of each diet (Figure 1A,B(1)).

After the 1-week acclimatization, the control group continued the standard diet, whereas the other special diet groups were switched to a 14% (*w*/*w*) hydrogenated coconut oil (HCO)-containing PUFA-deficient diet (AIN-93G based; Oriental Yeast) [17] for 5 weeks. The FA composition of this PUFA-deficient diet was as follows: C6:0, 0.2%; C8:0, 2.6%; C10:0, 6.4%; C12:0, 63.3%; C14:0, 15.7%; C16:0, 5.3%; and C18:0, 6.5% in total FA. To clarify the importance of PUFA supplementation, the PUFA (+) diet group, in addition to the PUFA-deficient diet, orally received 0.2 mL/mouse of *n*-3/*n*-6-balanced PUFA-containing oil (soybean oil [product number: 190-03776] plus linseed oil [product number: 125-01046] mixed at 1:1, *v*/*v*) by gavage every other day (typically at 7 pm) at a physiological level. To ensure the same calorie intake between the PUFA (+) diet and PUFA (−) diet groups, the PUFA (−) diet group orally received PUFA-deficient oil (coconut oil [product number: 036-20905]) by gavage as described previously [18]. The oil was purchased from Wako Pure Chemical Industries (Osaka, Japan).

To examine the difference in the response to PO treatment with each diet, these mouse groups underwent PO treatment from the third week after starting the corresponding diets (Figure 1B(2)). The PO treatment procedure was as follows: The mice received daily intraperitoneal injections of bovine serum albumin (BSA; Sigma Chemical, St. Louis, MO, USA) dissolved in saline for 1 or 2 weeks. The dose of BSA was gradually increased as follows: 0.1 g (1 day), 0.15 g (2 day), 0.2 g (3 day), 0.3 g (4 and 5 days), 0.35 g (6 and 7 days), and 0.4 g (8–14 days)/mouse. The mice injected with BSA for 1 week were defined as 1w PO control (*n* = 5), 1w PO PUFA (+) (*n* = 5), and 1w PO PUFA (−) (*n* = 5). Similarly, the mice injected with BSA for 2 weeks were defined as 2w PO control (*n* = 5), 2w PO PUFA (+) (*n* = 5), and 2w PO PUFA (−) (*n* = 5). After the experimental period, the mice were fasted overnight and anesthetized with isoflurane (Wako, Osaka, Japan). The kidneys were collected and decapsulated, and the cortex was separated for pathological and biochemical analyses. The collected cortex was stored at −80 °C until biochemical analysis.

Through the experiments, the sample size was determined by taking into consideration the minimum number required to obtain a statistically significant difference and accidental losses during the experiment.

### 2.3. Urinary Protein Analysis

Daily urine collection was performed 3 days before the commencement of the intraperitoneal injection of BSA. Furthermore, 24 h urine samples were collected daily and precipitated with 20% (*w*/*v*) trichloroacetic acid. After treatment as previously described [16], urinary proteins were measured using a bicinchoninic acid (BCA) protein assay kit (Thermo Fisher Scientific, Rockford, IL, USA).

### 2.4. Pathological Analysis

Mouse kidney cortex tissue slices were fixed in 10% formalin and embedded in paraffin. The sectioned samples were deparaffinized and stained with periodic acid–methenamine silver (PAM) for pathological analysis and observed using a light microscope. For immunofluorescence (IF)-based analysis, the deparaffinized sections of the kidney were subjected to antigen retrieval by 600 kw microwave in 0.01 M sodium citrate for 30 min, followed by permeabilization with 0.3% Triton X-100 in PBS for 30 min at room temperature. After blocking with 4% BSA for 60 min, the slides were immunostained overnight at 4 °C with antibodies against lysosome-associated membrane protein type 1 (Lamp1; 1:200 dilution; mouse monoclonal; American Research Products, Belmont, MA, USA). After washing thrice with PBS, the samples were incubated with FITC-conjugated secondary antibodies (Jackson ImmunoResearch, West Grove, PA, USA) at room temperature for 1 h. Subsequently, the samples were washed and stained with 4′,6-diamidino-2-phenylindole to visualize the nuclei and then mounted using a fluorescence mounting medium (Agilent, Santa Clara, CA, USA). The slides were observed using a Zeiss LSM880 confocal imaging system (Fluoview; Olympus, Tokyo, Japan). The tissues used for transmission electron microscopy (TEM) were sliced into approximately 1 mm^3^ cubes and fixed in 2.5% glutaraldehyde. The samples were then dehydrated and embedded in Epon resin. Ultrathin sections were double-stained with uranyl acetate and lead citrate and observed at 80 kV using a JEM 1400 electron microscope (JEOL, Tokyo, Japan) [19]. The number of lysosomes in 30 random TEM images of the renal proximal tubules was measured and compared between the groups.

### 2.5. mRNA Analysis

Total RNA was extracted from the renal cortex using an RNeasy Mini Kit (QIAGEN, Hilden, Germany) and a PrimeScript RT Reagent Kit (Takara Bio, Otsu, Japan). The cDNAs were quantified using NanoDrop 2000 (Thermo Fisher Scientific, Rockford, IL, USA). The gene expression of related mRNA was measured via quantitative real-time polymerase chain reaction (qPCR) using SYBR Premix Ex Taq II (Takara Bio) on a Step One Plus system (Thermo Fisher Scientific, Rockford, IL, USA). Specific primers were designed, as listed in Table 1. Data analysis was performed using the ΔΔCt method. After normalization to glyceraldehyde-3-phosphate dehydrogenase (GAPDH) gene levels, data were presented as fold changes, with the value of the control mice denoted as 1.

### 2.6. Immunoblot Analysis

The preparation of renal cortical tissue lysates and immunoblot analysis were performed as described previously [20]. Immunoblotting was performed using antibodies against long-chain acyl-CoA synthetase (LACS) [21], very long-chain acyl-CoA dehydrogenase (VLCAD) [22], long-chain acyl-CoA dehydrogenase (LCAD) [23], medium-chain acyl-CoA dehydrogenase (MCAD) [24], mitochondrial trifunctional protein α and β subunits (TPα and TPβ) [25], acyl-CoA oxidase 1 (AOX1) [26], peroxisomal bifunctional protein (PH) [27], peroxisomal thiolase (PT) [28], carnitine palmitoyl-CoA transferase (CPT II) [29], β-actin (Abcam, Cambridge, UK), microtubule-associated protein 1 light chain 3B (LC3B) (Novus Biologicals, Centennial, CO, USA), P62, Beclin1, and Atg5 (Medical & Biological Laboratories, Co., Ltd., Nagoya, Japan). After incubation with the primary antibodies, the membranes were incubated with the horseradish-peroxidase secondary antibodies and treated with ECL Prime Western blotting Detection Reagent (GE Healthcare, Little Chalfont, UK). Band intensities were measured densitometrically using an ECL Imager (Thermo Fisher Scientific, Rockford, IL, USA), compared with β-actin, and subsequently expressed as fold changes relative to those of the control mice.

### 2.7. Other Experiments

The DNA-binding activity of PPARα was determined using a PPARα Transcription Factor Assay kit (Cayman Chemical, Ann Arbor, MI, USA). Nuclear protein preparation from the renal cortex was carried out using the NE-PER nuclear and cytoplasmic extraction reagent (Thermo Fisher Scientific, Rockford, IL, USA) and quantified using a BCA protein assay kit (Thermo Fisher Scientific, Rockford, IL, USA). The results are expressed as fold changes relative to those of the control mice. For the measurement of kidney ATP content, 30 mg of kidney sample was homogenized in 600 μL ultrapure water and centrifuged (10,000× *g* at 4 °C for 10 min). The ATP concentration in the supernatant was measured using a Tissue ATP assay kit (Toyo B-Net, Tokyo, Japan) according to the manufacturer’s instructions.

### 2.8. Statistical Analysis

All the continuous variables were evaluated using the Shapiro–Wilk test for normality, and those that exhibited a normal distribution are presented as mean ± standard deviation. Intergroup differences were ascertained using the analysis of variance, followed by Tukey’s post hoc test, using the SPSS software v26J (IBM, New York, NY, USA). Statistical significance was set at *p* < 0.05.

## 3. Results

### 3.1. PUFA-Deficient Diet Significantly Increased Urinary Protein Excretion Following PO

During the experiment, there was no significant difference in food intake among the groups, and all the mice survived. There were no remarkable changes in body weight and organ weight among the control, PUFA (+), and PUFA (−) groups at any time point (Table 2).

To investigate the effect of the special diet on urinary protein excretion (without PO), the time-course changes in daily urinary protein excretion were examined in the control, PUFA (+), and PUFA (−) groups. There was no significant intergroup difference in daily urinary protein excretion during the experimental period. Under the PO experimental conditions, the amount of daily urinary protein excretion increased in a time-dependent manner in all the diet groups, which indicated an adaptive urinary protein excretion against PO (Figure 2). Compared with the PO control group, a significantly larger increase in daily urinary protein excretion was observed in the PO PUFA (−) group, especially from the second week after the commencement of BSA injections. Excess urinary protein excretion was attenuated by PUFA supplementation at the physiological intake level in the PO PUFA (+) group. These findings suggest that urinary protein excretion under normal conditions is not affected by the dietary type, but rather that PUFA deficiency considerably influences the adaptive urinary protein excretion process secondary to PO. PUFA supplementation at the physiological intake level attenuated this unusual adaptive response, and this highlights the importance of PUFA deficiency.

### 3.2. PUFA-Deficient Diet Increased Lysosomes and Suppressed the PO-Induced Adaptive Endocytosis Activation

To prevent protein loss from urine, urinary protein is reabsorbed in the renal proximal tubules through receptor-mediated endocytosis, and protein reabsorption endocytic vesicles are found close to the brush border in proximal tubular cells [30]. Pathological analyses were performed to determine the cause of the abnormal increase in urinary protein excretion in the PO PUFA (−) group (Figure 3A).

In the absence of PO, light microscopy-based analysis using PAM staining showed multiple small dark-brown vesicular structures near the brush border, which were suspected to be endocytic vesicles to a similar extent in each diet group. Furthermore, this analysis detected multiple microvacuoles that were larger than the small dark-brown vesicular structures in the PUFA (−) group.

In the 2w PO experimental conditions, the dark-brown small vesicular structures were markedly increased in the proximal tubules in the 2w PO control and 2w PO PUFA (+) groups, suggesting that the activation of adaptive endocytosis occurred in response to increased urinary protein levels. In contrast, the PO-induced increase in the dark-brown small vesicular structures was scarcely observed in the 2w PO PUFA (−) group, whereas multiple microvacuoles were detected. No glomerular abnormalities were detected in any of the groups, indicating a lower contribution of glomeruli to phenotypic changes (Figure 3B).

As an earlier study demonstrated that a PUFA-deficient diet markedly increased lysosomes in PTECs [18], IF analysis of the lysosomal membrane marker Lamp1 and TEM analysis were conducted. The IF analysis showed many Lamp1-positive vesicles in proximal tubules of the PUFA (−) and 2w PO PUFA (−) groups (Figure 4), whereas these were scarcely detected in the other groups.

The TEM analysis revealed a marked increase in lysosomes in the proximal tubules of the PUFA (−) and 2w PO PUFA (−) groups (Figure 5A). The quantitative testing of lysosome numbers from 30 random TEM images demonstrated that compared to those in the 2wPO control group, lysosomes were significantly increased in the 2wPO PUFA (−) group. Therefore, the multiple microvacuoles found in the PTECs of the PUFA (−) and 2w PO PUFA (−) groups were speculated to be lysosomes. High-magnification TEM revealed many lysosomes containing various components that could be autophagolysosomes (Figure 5B). These findings suggested that a PUFA-deficient diet increases the number of lysosomes and suppresses adaptive endocytosis in response to PO.

### 3.3. PUFA-Deficient Diet Induced Renal Continuous Autophagy Activation, Possibly Contributing to Lysosomal Increase

A previous study demonstrated that the PUFA deficiency-induced renal lysosomal increase was attributable to the activation of autophagy [18]; additionally, the study revealed the molecular mechanism of autophagy activation caused by PUFA deficiency. Autophagy is activated by the suppression of the mammalian target of rapamycin (mTOR), and AKT or the mitogen-activated protein kinase (MAPK)/extracellular regulated protein kinase-1/2 (Erk1/2) signaling cascade upstream of mTOR are the main regulatory pathways to suppress autophagy. The study demonstrated that PUFA deficiency reduced Erk1/2 phosphorylation levels and remarkably inhibited mTOR phosphorylation, suggesting that the Erk/mTOR pathway is the main signaling cascade for the induction of autophagy and lysosomal increase induced by a PUFA-deficient diet. Another earlier study using starved *Ppara^−/−^* mice showed excess urinary protein excretion and a marked increase in lysosomes [16]. These lysosomes contained a large amount of undigested material and formed aggregated giant lysosomes. The authors concluded that renal energy depletion caused the stagnation of endocytosis–lysosomal flux, resulting in increased lysosome numbers, which impaired urinary protein reabsorption. As both endocytic and autophagic vesicles enter the same phagosome–lysosome flux, based on previous reports, we speculated that autophagy–lysosome flux potentially interfered with endocytosis–lysosome flux, and consequently, resulted in the disruption of urinary protein reabsorption that was demonstrated in the current study. Therefore, we evaluated the mRNA expression levels of the autophagy-related protein Beclin1, lysosomal marker Lamp1, and representative lysosomal enzyme *β*-glucuronidase, and the protein expression levels of the autophagy marker LC3B, autophagy-related proteins Atg5 and Beclin1, and autophagy substrate P62 in each group.

Compared with the 2w PO control group, the mRNA expression of Lamp1 and *β*-glucuronidase significantly increased in the 2w PO PUFA (−) group (Figure 6A). In the absence of PO, the mRNA expression of these factors did not differ among the diet groups; however, the protein expression levels of LC3B, Atg5, and Beclin1 were significantly increased whereas that of P62 was significantly decreased in the PUFA (−) group (Figure 6B), which indicated the activation of autophagy. These autophagy-activation responses seemed to continue in the 2w PO PUFA (−) group. The changes in mRNA and protein expression were attenuated by PUFA supplementation at the physiological intake level in the PUFA (+) group, which suggests the importance of PUFA deficiency in autophagy activation. These findings suggest that a PUFA-deficient diet causes the continuous activation of renal autophagy and possibly contributes to an increase in lysosome numbers.

### 3.4. PUFA-Deficient Diet Significantly Aggravated PO-Induced Impaired FA Metabolism and PPARα Function and Induced the Disruption of Renal Energy Homeostasis

A previous study demonstrated that PO induces FA overload in PTECs via BSA-binding FA, exerts FA toxicity, decreases the ability for FA β-oxidation and PPARα, and disrupts energy homeostasis [31]. Furthermore, studies in *Ppara^−/−^* mice demonstrated that PPARα prevents PO-induced FA toxicity, and maintenance of FA β-oxidation ability is essential for energy homeostasis and urinary protein-reabsorptive ability in PTECs [16,19]. Therefore, we examined the expression levels of FA β-oxidation-related enzymes, PPARα activity, and the amount of renal ATP. Figure 7 shows the mRNA expression levels of representative FA β-oxidation-related enzymes, including VLCAD, CPT II, PH, PT, TPα, and TPβ, in the renal cortex of each group. In the absence of PO, none of the diets had a significant effect on the mRNA expression of these enzymes (Figure 7). Under the PO experimental condition, PO time-dependently decreased the mRNA expression of these β-oxidation-related enzymes. The mRNA expression of most of these enzymes in the 2w PO PUFA (−) group was significantly lower than that in the 2w PO control group. These abnormal decreases were attenuated by PUFA supplementation at the physiological intake level in the 2w PO PUFA (+) group. Moreover, immunoblot analyses of the protein expression of 10 β-oxidation-related enzymes were conducted (Figure 8). The protein expression of most of these enzymes was significantly lower in the 2w PO PUFA (−) group than in the 2w PO control group, and PUFA supplementation attenuated these protein reductions, which was consistent with the results of the mRNA analyses.

As the changes in the PPARα activity and renal energy could contribute to the current experimental model, the PPARα DNA-binding activity and amount of ATP in the renal cortex were examined in each group. In the experimental condition without PO, the PPARα DNA-binding activity and amount of renal ATP did not differ among the diet groups (Figure 9). In the PO experimental condition, both the PPARα DNA-binding activity and the amount of ATP in the 2w PO PUFA (−) group were significantly lower than those in the 2w PO control group, and PUFA supplementation at the physiological intake level attenuated these reductions.

These findings suggest that a PUFA-deficient diet significantly aggravates the diminished abilities of FA metabolism and PPARα induced by PO and disrupts energy homeostasis.

## 4. Discussion

This study revealed that with a PUFA-deficient diet, PO caused various renal changes, an abnormal increase in daily urinary protein excretion, an increase in the number of lysosomes, and the suppression of adaptive endocytosis activation, probably via continuous autophagy activation, and aggravated the reduction in FA metabolism and PPARα activity secondary to PO, with the consequent disruption of renal energy homeostasis. However, these changes were attenuated by PUFA supplementation at the physiological intake level. These findings suggest that PUFAs are essential nutrients for this adaptive response to excess urinary protein excretion.

Figure 10 shows a schematic representation of the speculated mechanism explaining the above phenomena and the importance of PUFAs for urinary protein reabsorption in PTECs during PO.

In general, urinary PO stress impairs PPARα function and FA metabolism (Figure 10A). As PUFAs are natural ligands of PPARα, the normal intake of PUFAs supports the abilities of PPARα and FA metabolism, with the resultant maintenance of renal energy homeostasis. Sufficient energy supply facilitates important proximal tubular functions requiring ATP: urinary protein reabsorption via endocytosis, lysosome maturation, and the digestion of absorbed protein. Eventually, the PO-induced increase in urinary protein excretion was attenuated.

On the one hand, PUFA deficiency can potentially impair PPARα function and FA metabolism, leading to reduced ATP production (Figure 10B). Furthermore, PUFA deficiency and ATP reduction cause continuous autophagy activation in PTECs. Autophagy plays dual pathogenic roles in kidney disease [32,33]. During acute-phase kidney injury, autophagy is initiated in the proximal tubules as a protective response. However, in the recovery phase, sustained autophagy activation can induce phenotypic transformations in the proximal tubule cells, leading to inadequate repair, interstitial fibrosis, and an accelerated transition from acute to chronic kidney disease [34]. On the other hand, autophagy prevents renal fibrosis by regulating excessive intracellular collagen degradation [35]. However, several previous studies have demonstrated that autophagy activation cannot continue to increase, reach saturation, or eventually weaken, even with continuous autophagy stimulation signals such as angiotensin II and H_2_O_2_ [36,37]. Continuous autophagy activation induced by various situations causes the stagnation of the autophagy–lysosomal flux, impeding lysosome maturation and the digestion of lysosomal contents, and may lead to renal tubular atrophy and the promotion of renal fibrosis [38,39,40]. Furthermore, a close relationship between autophagy and FA metabolism has been reported [41,42]. Thus, autophagy can activate PPARα, whereas insufficient autophagy activity suppresses PPARα function and FA metabolic ability, contributing to reduced energy production [43,44]. In contrast, PPARα enhances autophagy by increasing the expression of autophagy-related genes [45,46]. As observed in the current study, the continuous autophagy activation due to PUFA deficiency and ATP reduction might impair the autophagy-lysosomal flux, contributing to further inactivation of PPARα and FA metabolism, followed by the aggravation of insufficient autophagy function. The energy disruption caused by this vicious cycle increases the number of immature lysosomes containing undigested proteins, which interferes with the adaptive activation of the endocytosis–lysosomal flux, resulting in an increase in urinary protein excretion. Although the above hypothesis was not fully proven, the current study indicates, for the first time, the pathological mechanism and underscores the importance of PUFAs in kidney diseases with urinary protein excretion.

Proximal tubules are important nephrons that utilize large amounts of ATP to actively reabsorb various urine substrates, including proteins, salt, electrolytes, sugar, and uric acid, which are important for maintaining body fluid homeostasis [47]. The current study suggested that PUFAs support adaptive tubular protein reabsorption by maintaining PPARα function, FA metabolism, and energy homeostasis in high-demand conditions. However, this continuous adaptive process consumes large amounts of energy and leads to proximal tubule overwork. If this tubular stress continues without energy to maintain the internal system, tubular cells may become damaged, progressing to tubular atrophy and interstitial fibrosis. As tubular atrophy and interstitial fibrosis are more closely associated with the progression of chronic kidney disease rather than glomerular damage [48,49], their prevention is crucial. Recently, the renoprotective effects of sodium–glucose cotransporter 2 inhibitors have attracted attention. They suppress the reabsorption of salt and sugars in PTECs, thereby reducing tubular overwork and renal energy demand, and exerting tubular protective effects, one of their proposed mechanisms of renal protection [50]. This clinical evidence suggests that reducing renal tubular stress through excess reabsorption and maintaining renal energy homeostasis are important interventions for renal protection. The current study showed that PUFA deficiency worsens the adaptive response of renal proximal tubules to proteinuria; however, this tubular adaptive response stress can be improved by maintaining renal energy homeostasis by PUFA supplementation at the physiological intake level. This demonstrates the importance of appropriate PUFA intake for normal tubular function. Epidemiological studies have reported that circulating PUFAs contribute to the maintenance of renal function, and PUFA deficiency has been observed in patients with chronic kidney disease [51,52]. The current study’s results suggest that appropriate PUFA intake is necessary in patients with kidney disease-associated proteinuria; however, this study did not examine the usefulness of PUFA supplementation beyond the physiological intake level. To verify the effects of higher PUFA supplementation, studies with other experimental designs would be necessary.

Previous reports indicate that PUFA supplementation beyond the physiological intake level confers favorable clinical effects in various kidney diseases via mechanisms other than the improvement of PUFA deficiency. Dietary *n*-3 PUFA confers renoprotection in mice with immune- and/or inflammation-mediated renal disorders [53], and *n*-3 fatty acid attenuates kidney fibrosis via AMPK-mediated autophagy activation [54]. In clinical settings, *n*-3 PUFA supplementation has been reported to have therapeutic potential in reducing proteinuria and kidney dysfunction in IgA nephropathy and other chronic kidney diseases and in reducing inflammation in patients undergoing dialysis [55,56]. The supplementation of *n*-3 PUFAs appears to diminish nephrotoxicity and hypertensive complications by inhibiting inflammatory and atherogenic mechanisms in lupus nephritis [57]. Dietary PUFAs have been reported to exert various biological functions such as improving dyslipidemia, mediating renal prostaglandin production rearrangement, decreasing the production of pro-inflammatory leukotrienes, maintaining endothelial function, and controlling the escape rate of albumin across capillaries [58]. Besides the above clinical effects, dietary PUFAs may protect PTECs via renal energy maintenance effects, which were observed in this study, and may contribute to the favorable clinical effects. Therefore, active PUFA intake may be necessary to prevent the progression of kidney disease. However, the appropriate level of PUFA supplementation for patients with kidney disease-associated proteinuria remains unelucidated. Excessive intake of *n*-6 PUFAs can lead to dyslipidemia, including increased low-density lipoprotein (LDL) levels and decreased high-density lipoprotein (HDL) levels, LDL oxidation, and inflammatory stimulation owing to arachidonic acid pathway activation, which may increase the risk of cardiovascular diseases (CVDs) [59]. Excessive *n*-3 PUFA intake induces bleeding tendencies and excessive immunosuppression [60]. Additionally, excessive *n*-6 PUFA intake is particularly problematic, and a high *n*-6/*n*-3 ratio reportedly correlates with CVDs. In general, appropriate PUFA intake and a balanced *n*-6/*n*-3 ratio (<4:1) are necessary for maintaining health [61]. As excessive or imbalanced PUFA intake may induce adverse events, achieving an optimal balance is crucial. However, the appropriate PUFA intake level and PUFA balance for patients with kidney disease remain unelucidated, warranting further research.

PUFA deficiency rarely causes clinical problems in humans who can consume diverse diets, which include vegetable oils and fish that provide adequate PUFAs. However, PUFA deficiencies may occur in individuals who consume extremely low-fat diets, those with anorexia, and those who undertake long-term stringent dietary restrictions, such as in some monasteries and religious institutions. PUFA deficiencies can be induced by digestive diseases that cause fat malabsorption (e.g., inflammatory bowel disease, chronic pancreatitis with exocrine insufficiency, liver cirrhosis, and bile secretion insufficiency) and by iatrogenic factors, such as long-term dependence on enteral or central intravenous nutrition with inadequate lipid supply [62]. Furthermore, PUFA deficiency can occur in geographical regions such as sub-Saharan Africa (Ethiopia, Somalia), certain parts of South Asia (e.g., Bangladesh, Nepal, the Himalayan region of Tibet, Bhutan, Sri Lanka, and the rural areas of India), and war- and conflict-affected regions (e.g., Syria and Yemen), which are afflicted by malnutrition and low fat intake owing to chronic extreme food shortages, poverty, and a diet that predominantly comprises carbohydrates [63,64,65]. Under these special conditions, renal tubular functional maladaptation, as elucidated in this study, may occur. Thus, kidney disease is a common disease that can occur in individuals in various regions and across diverse dietary environments. Understanding the effects of PUFA deficiency on renal tubular function is important to elucidate the pathology of different patients with kidney diseases.

### Limitations

The current study has some limitations. First, we focused on the endocytosis–lysosomal degradation system, which is a mechanism of urinary protein reabsorption. Therefore, the effect of PUFAs on another reabsorptive mechanism that is mediated by Na^+^-K^+^-ATPase, which is an important urinary substrate reabsorption mechanism that requires a large amount of ATP, was not verified. When high concentrations of salt are ingested, or in cases of lifestyle-related diseases such as diabetes and hyperuricemia, renal tubules overwork and consume large amounts of ATP for reabsorption [66,67]. In such cases, dietary PUFAs may exert renoprotective effects. Further investigation is needed to determine the importance of PUFAs when PTECs are overloaded with other important tubular substrates, such as salt and sugar. Second, we speculated that the phagosome–lysosome flux may have stagnated in the PO PUFA-deficient diet group, secondary to phenomena such as continuous autophagy activation, increased lysosomes, decreased adaptive endocytosis activation, and decreased protein reabsorption capacity; however, these findings are not conclusive. To verify the stagnation of the phagosome–lysosome flux, further verification using the LC3 turnover assay with inhibitors such as bafilomycin A1 or chloroquine is required. Third, obvious pathological changes such as tubular cell damage and/or interstitial fibrosis were not observed in the current study; therefore, whether PUFAs can prevent or alleviate the pathological changes caused by PO remains unknown. Acute tubular necrosis, termed PO nephropathy, induces acute kidney injury [68,69,70]. In the current study, the amount of PO was gradually increased to examine the effect of PUFAs on adaptive urinary protein reabsorption in PTECs without obvious proximal tubular damage. Further studies under different experimental conditions are required to verify the effects of PUFAs on PO nephropathy.

## 5. Conclusions

Dietary PUFAs support FA metabolism and PPARα function in proximal tubules, suppress continuous excessive autophagy activation, and maintain tubular energy homeostasis. These beneficial effects of PUFAs contribute to adaptive endocytosis activation, which is an important tubular protein reabsorption process that protects against urinary PO and results in reduced urinary protein excretion. Thus, PUFAs are essential nutrients for maintaining renal tubular function against urinary protein stress. Active PUFA intake may be important for patients with kidney disease-associated proteinuria, especially for those with various PUFA deficiency-inducing conditions such as extremely low-fat diets; anorexia; long-term strict dietary restrictions; fat-malabsorptive digestive diseases; malnutrition and low fat intake owing to food shortages, poverty, and war and conflict; and a diet predominantly comprising carbohydrates.

## Figures and Tables

**Figure 1 nutrients-17-00961-f001:**
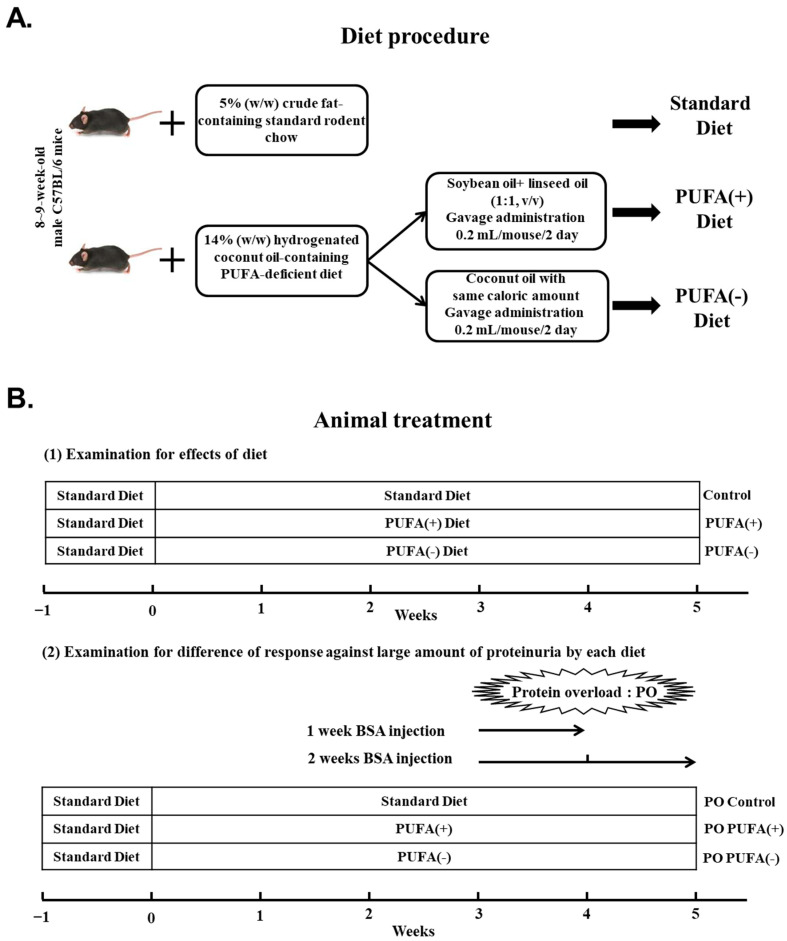
Dietary procedure and treatment of mice. (**A**) After the 1-week acclimatization, the control group was fed a 5% (*w*/*w*) crude fat-containing standard diet continuously, whereas the other diet groups were switched to a 14% (*w*/*w*) hydrogenated coconut oil (HCO)-containing polyunsaturated fatty acid (PUFA)-deficient diet for 5 weeks. The PUFA (+) diet group, in addition to the PUFA-deficient diet, was orally treated with PUFA-containing oil at a physiological level. To ensure equal calorie intake, the PUFA (−) diet group was orally administered the PUFA-deficient oil with the same caloric content. (**B**) (1) Examination of the effects of diet. After the 1-week acclimatization, the mice were randomly divided into the control (standard diet: *n* = 4), PUFA (+) diet (PUFA supplementation on PUFA-deficient diet: *n* = 4), and PUFA (−) diet (continuous PUFA-deficient diet: *n* = 4) groups. (2) Examination for differences in the responses to excessive protein overload (PO) in each dietary group. These mice received PO treatment from the third week after the initiation of the specific diet. These mice received daily intraperitoneal injections of bovine serum albumin solution for 1 week [1w PO control, 1w PO PUFA (+), and 1w PO PUFA (−)] or 2 weeks [2w PO control, 2w PO PUFA (+), and 2w PO PUFA (−)] (each group, *n* = 5).

**Figure 2 nutrients-17-00961-f002:**
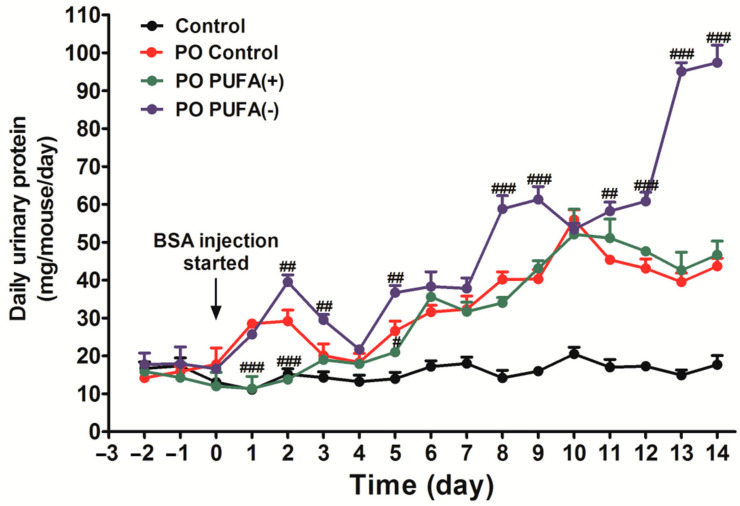
Diet-stratified differences in daily urinary protein excretion in protein overload (PO) mice. Daily total urinary protein excretion in each mouse group (mg/mouse/day) was compared. Black line, control (standard diet without PO); red line, PO control (standard diet with PO); green line, PO PUFA (+); blue line, PO PUFA (−) groups. Data are expressed as the mean ± SD. Statistically significant differences: ^#^
*p* < 0.05, ^##^
*p* < 0.01, and ^###^
*p* < 0.001 vs. the PO control group.

**Figure 3 nutrients-17-00961-f003:**
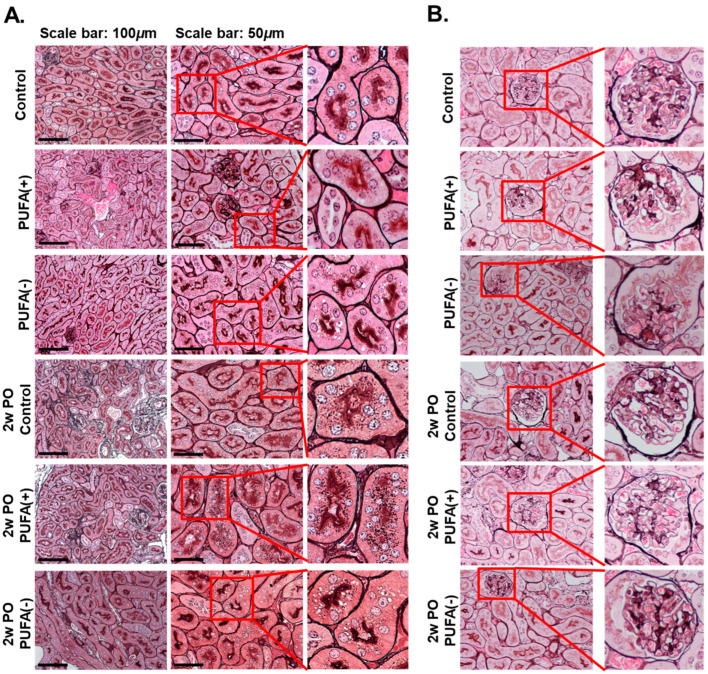
Pathological analyses of kidney tissues. (**A**) Light microscopic analysis. The kidney tissue sections were stained with periodic acid–methenamine silver (PAM). The scale bars in the left panels and the middle panels represent 100 and 50 μm, respectively. The right panels show enlarged images of the square area in the middle panels. (**B**) Pathological analyses for glomeruli. Representative glomeruli from each group are shown. The right panels show enlarged images of the square areas in the left panels.

**Figure 4 nutrients-17-00961-f004:**
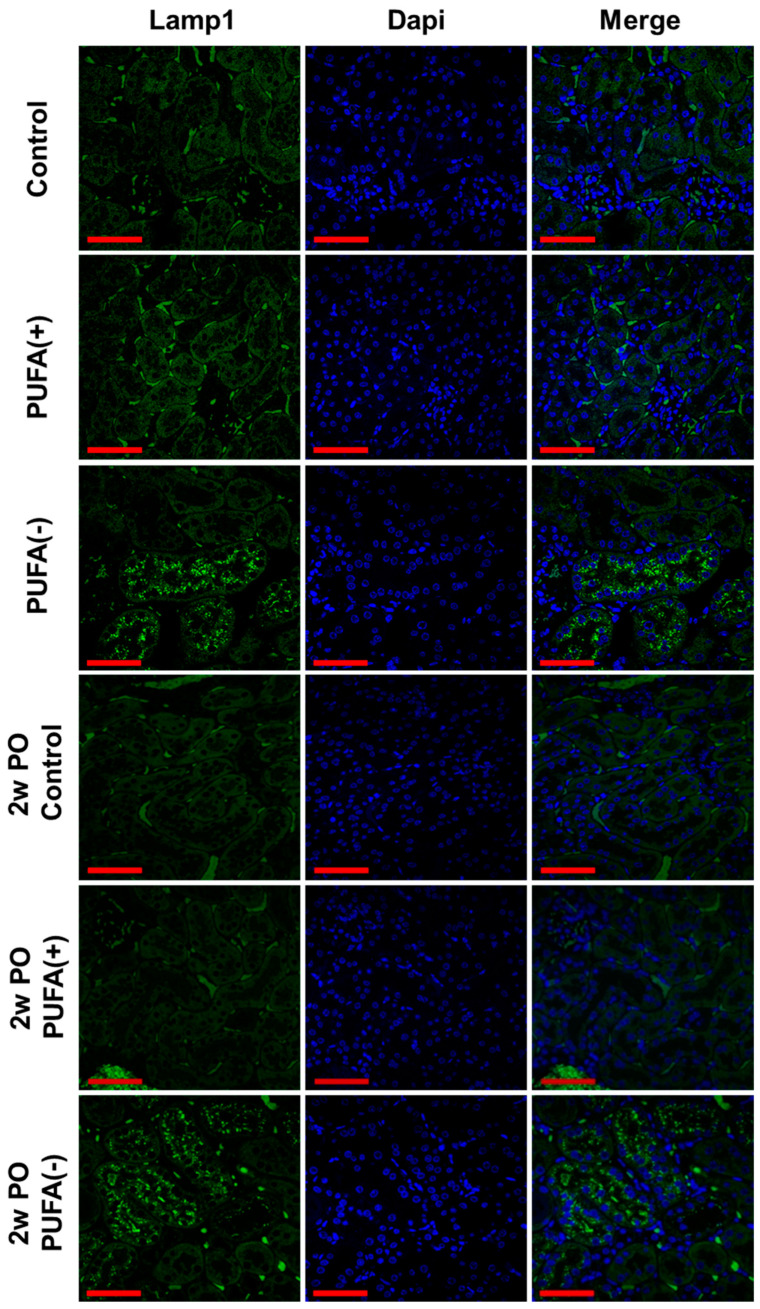
Immunofluorescence staining of lysosomal membrane marker, Lamp1, in the renal cortex. Confocal images of kidney samples obtained from each group. Lamp1 was visualized using a FITC-conjugated secondary antibody (green), followed by counterstaining with DAPI (blue). Scale bar = 50 μm.

**Figure 5 nutrients-17-00961-f005:**
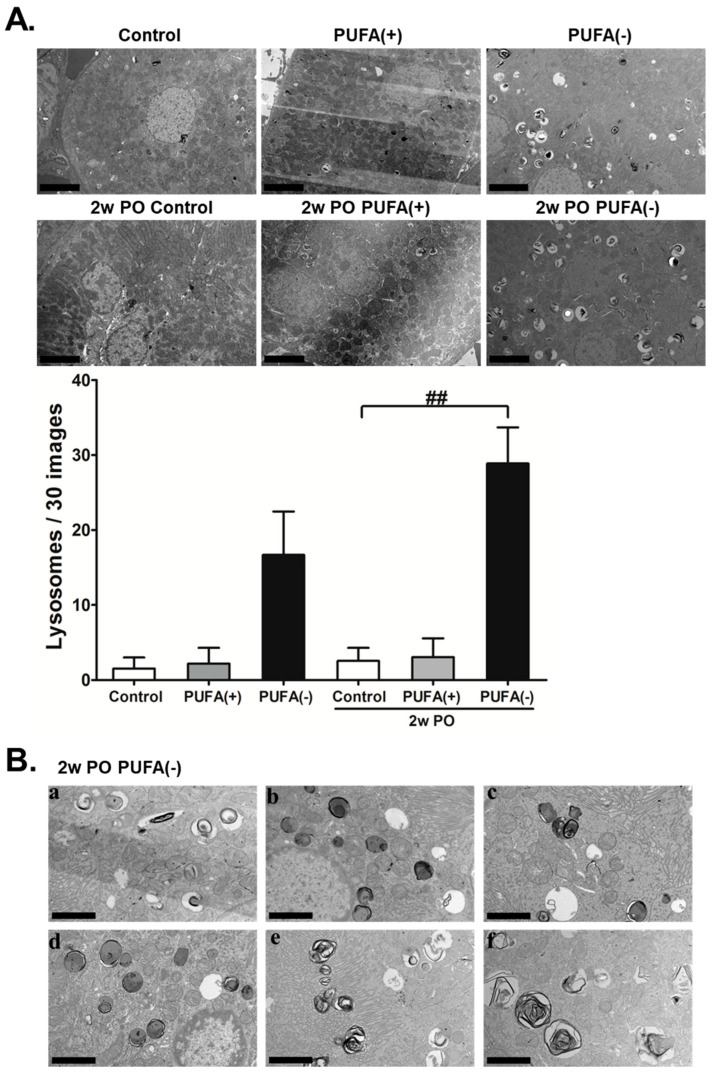
Transmission electron microscopy (TEM) analyses of kidney tissues. (**A**) Representative electron micrographs of sections containing proximal tubules. The number of lysosomes counted in 30 random TEM images of each mice group are indicated. Scale bar = 5 μm. Data are expressed as the mean ± SD. Statistically significant differences: ^##^
*p* < 0.01 vs. the 2w PO control group. (**B**) TEM analysis under high magnification in the 2w PO PUFA (−) group. (**a**–**f**) Lysosomes containing various components were observed. Scale bar = 2.5 μm.

**Figure 6 nutrients-17-00961-f006:**
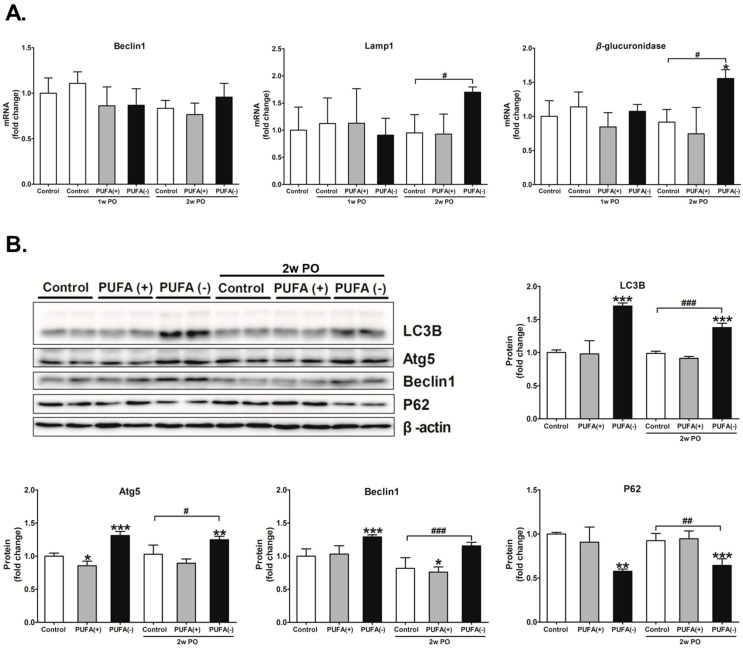
Renal expression of autophagic and lysosomal related enzymes. (**A**) mRNA expression of the autophagy-related protein Beclin1, lysosomal marker Lamp1, and representative lysosomal enzyme *β*-glucuronidase in each mouse group. The mRNA levels of these enzymes were analyzed via real-time PCR and normalized to that of GAPDH. (**B**) Protein expression of autophagy marker LC3B, autophagy-related proteins Atg5 and Beclin1, and autophagy substrate P62. The protein levels were normalized to that of β-actin. All the data are shown as fold changes relative to the control group. Data are expressed as the mean ± SD. The white, gray, and black bars indicate the control, PUFA (+), and PUFA (−) groups, respectively. Statistically significant differences: * *p* < 0.05, ** *p* < 0.01, and *** *p* < 0.001 vs. the control group; ^#^ *p* < 0.05, ^##^
*p* < 0.01, and ^###^ *p* < 0.001 vs. the 2w PO control group.

**Figure 7 nutrients-17-00961-f007:**
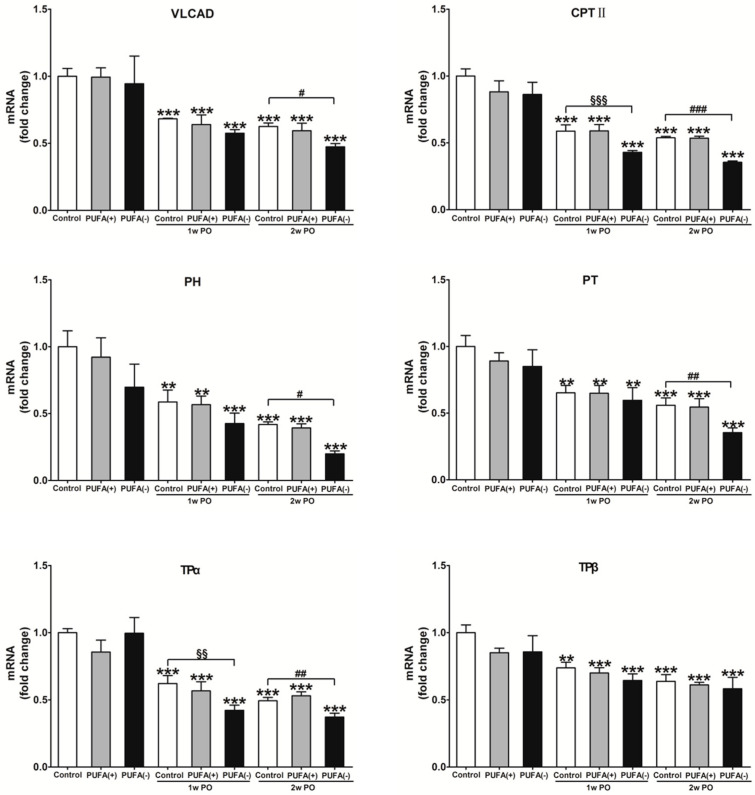
Renal mRNA expression of major enzymes in the mitochondrial and peroxisomal β-oxidation system. CPT II acts on fatty acid transport from the cytosol to mitochondria. VLCAD, TPα, and TPβ catalyze the β-oxidation reaction of long-chain fatty acids. PH and PT are mainly responsible for the dehydration, dehydrogenation, and sulfidation processes in peroxisomes. The mRNA levels of these enzymes were analyzed via real-time PCR and normalized to those of GAPDH. All the data are shown as fold changes relative to the control group and expressed as the mean ± SD. The white, gray, and black bars indicate the control, PUFA (+), and PUFA (−) groups, respectively. Statistically significant differences: ** *p* < 0.01 and *** *p* < 0.001 vs. the control group; ^§§^ *p* < 0.01 and ^§§§^ *p* < 0.001 vs. the 1w PO control group; ^#^ *p* < 0.05, ^##^ *p* < 0.01, and ^###^ *p* < 0.001 vs. the 2w PO control group.

**Figure 8 nutrients-17-00961-f008:**
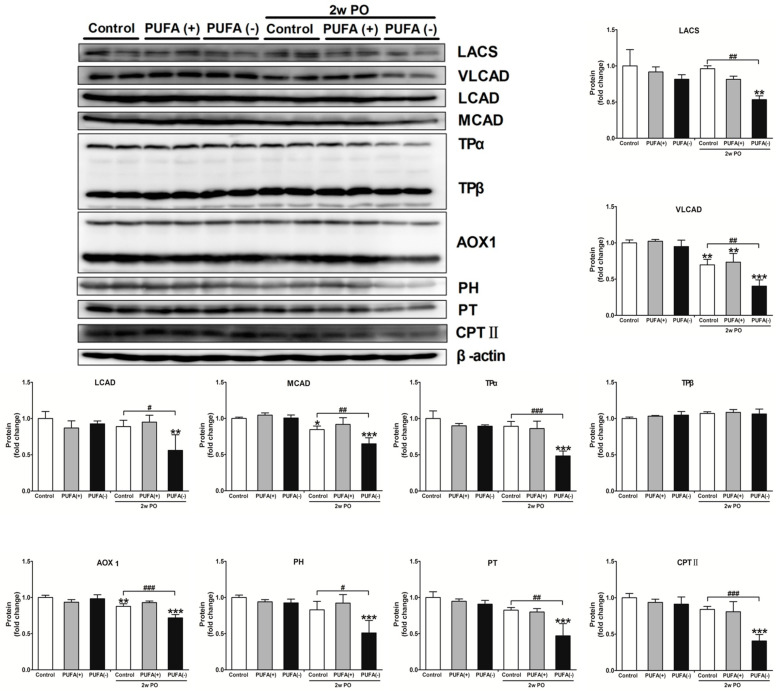
Immunoblot analyses of fatty acid β-oxidation-metabolizing enzymes in the renal cortex. Immunoblot analyses showing the protein expression levels of the mitochondrial β-oxidation enzymes (LACS, VLCAD, LCAD, MCAD, TPα, and TPβ), peroxisomal β-oxidation enzymes (AOX1, PH, and PT), and CPT II. The protein levels were normalized to that of β-actin. All the data are shown as fold changes relative to the control mice. Data are expressed as the mean ± SD. White, gray, and black bars indicate control, PUFA (+), and PUFA (−) groups, respectively. Statistically significant differences: * *p* < 0.05, ** *p* < 0.01, and *** *p* < 0.001 vs. the control group; ^#^ *p* < 0.05, ^##^ *p* < 0.01, and ^###^ *p* < 0.001 vs. the 2w PO control group.

**Figure 9 nutrients-17-00961-f009:**
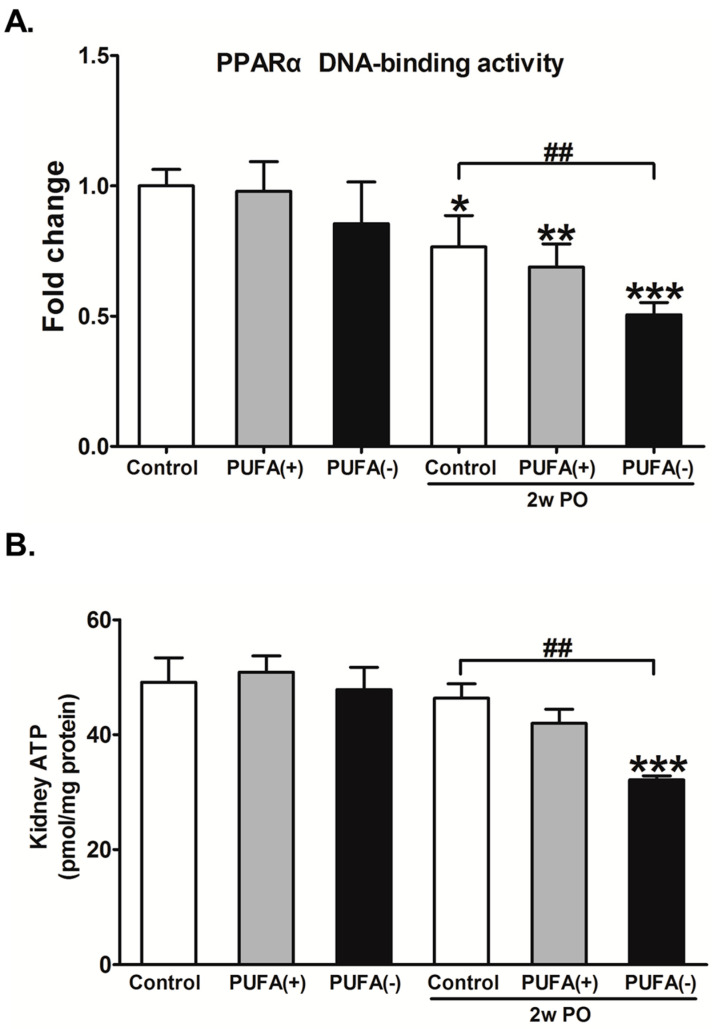
Effects of polyunsaturated fatty acid (PUFA)-deficient diet on peroxisome proliferator-activated receptor α (PPARα) DNA-binding activity and ATP level in the renal cortex. (**A**) The DNA-binding activity of PPARα in the kidney cortex. The data are shown as fold changes relative to the control mice. (**B**) The ATP level in the kidney cortex. All data are expressed as the mean ± SD. White, gray, and black bars indicate control, PUFA (+), and PUFA (−) groups, respectively. Statistically significant differences: * *p* < 0.05, ** *p* < 0.01, and *** *p* < 0.001 vs. the control group; ^##^ *p* < 0.01 vs. the 2w PO control group.

**Figure 10 nutrients-17-00961-f010:**
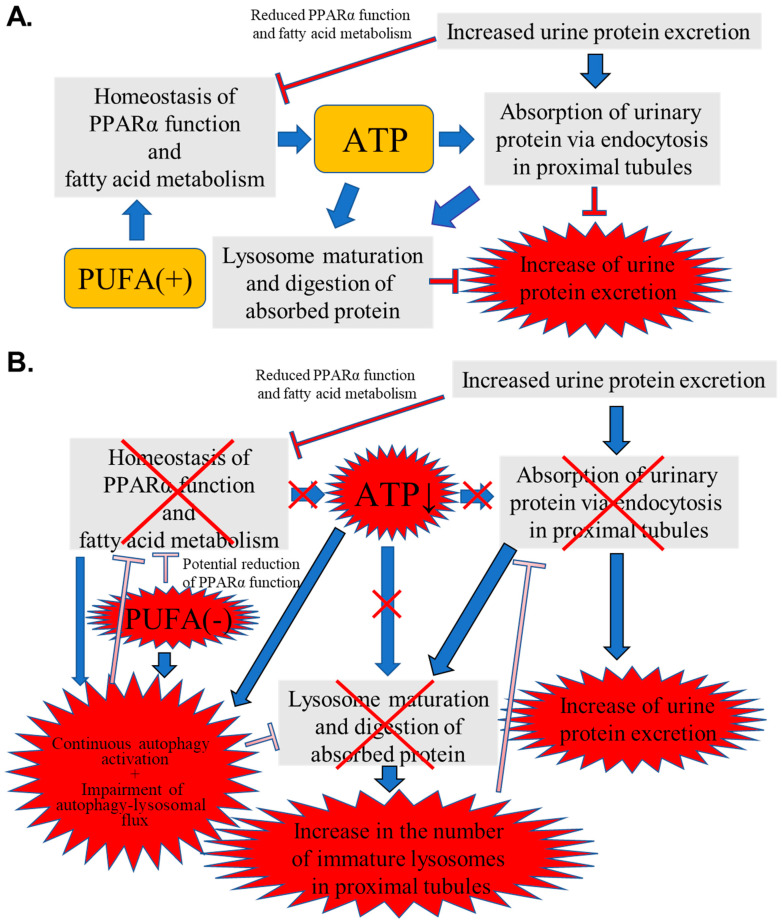
Schematic representation of the speculated mechanism explaining the observed phenomena and importance of polyunsaturated fatty acids (PUFAs) for urinary protein reabsorption in proximal tubular epithelial cells (PTECs) during protein overload (PO). (**A**) Normal intake of PUFAs. (**B**) PUFA-deficient status. Red X means impairment of function.

**Table 1 nutrients-17-00961-t001:** Primer pairs used for qPCR.

Gene	Accession Number	Primer Sequence (5′ to 3′)
Acadvl (VLCAD)	NM_017366	F GCGTGTGCTCCGAGATATTC
		R CCAGTGAGTTCCTTTCCTTTG
CPT II	NM_009949	F ATCGTACCCACCATGCACTAC
		R CTGTCATTCAAGAGAGGCTTCTG
Acaa1 (PT)	NM_130864	F TCTACGGTCAACAGACAGTGTTCA
		R GGCCATGCCAATGTCATAAGA
PH	NM_023737	F CGATACTCTTCCCCCACTACCA
		R CAGTTACCAACAACGACTCCAATC
HADHA (TPα)	NM_178878	F CCTTTATCCTGCCCCTTTG
		R GCGATTCAGCAAGATAACCA
HADHB (TPβ)	NM_145558	F AGCGCCTGTCCTTACTCAGT
		R CATGGTCTCATTAGTGGAGAACTC
Lamp1	NM_010684	F ACACTGCACACAGGATGGAC
		R CTCTGGTCACCGTCTTGTTGT
Gusb (*β*-glucuronidase)	NM_010368	F GCAGCCCTTCGGGACTTTAT
		R CCCATTCACCCACACAACTG
Becn1 (beclin 1)	NM_019584	F GGCTAACTCAGGAGAGGAGC
		R ATCAGATGCCTCCCCGATCA
GAPDH	M32599	F TGCACCACCAACTGCTTAG
		R GGATGCAGGGATGATGTTCTG

F, forward sequence; R, reverse sequence.

**Table 2 nutrients-17-00961-t002:** Changes in body and organ weight.

	Body Weight (g)	Body Weight Change (%)	Kidney/Body Weight (%)	Liver/Body Weight (%)
Control	25.1 ± 0.8	nd	1.3 ± 0.1	4.0 ± 0.4
PUFA (+)	27.8 ± 0.3	109.7 ± 3.5	1.4 ± 0.1	4.4 ± 0.6
PUFA (−)	26.6 ± 1.7	105.1 ± 9.2	1.3 ± 0.1	4.6 ± 0.3
1w PON Control	27.5 ± 1.6	125.6 ± 10.0	1.4 ± 0.1	4.6 ± 0.1
1w PON PUFA (+)	26.5 ± 1.5	121.1 ± 10.0	1.5 ± 0.2	4.3 ± 0.8
1w PON PUFA (−)	28.0 ± 0.7	128.0 ± 8.4	1.4 ± 0.1	4.2 ± 0.3
2w PON Control	26.9 ± 1.0	123.2 ± 10.7	1.3 ± 0.1	4.4 ± 0.2
2w PON PUFA (+)	27.1 ± 1.0	124.0 ± 10.1	1.5 ± 0.1	4.4 ± 0.5
2w PON PUFA (−)	27.0 ± 1.1	123.2 ± 5.8	1.5 ± 0.2	4.5 ± 0.4

Test mice were fed a 13.5% hydrogenated coconut oil-containing PUFA-deficient diet for 5 weeks. The control group was fed 5% (*w*/*w*) crude fat-containing standard rodent chow for comparison. The data are shown as the mean ± SD. nd, not determined. There were no significant differences between the PUFA (+) and PUFA (−) groups at any time point.

## Data Availability

The datasets generated and/or analyzed in the current study are available from the corresponding author upon reasonable request due to data management.

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
