# Peer review of "Dietary Polyunsaturated Fatty Acid Deficiency Impairs Renal Lipid Metabolism and Adaptive Response to Proteinuria in Murine Renal Tubules"

_nutrients, 2025, doi:10.3390/nu17060961_

Round 1

Reviewer 1 Report

Comments and Suggestions for Authors

The paper is interesting. This field of research actually deserves our attention also to reduce the impact of nephropathies. Methodology and statistycal tools are clearly described and the tests employied are simple. I agree with the limitations described by Authors. Further studies are needed to clarify the role of PUFA supplementation in humans with kidney diseases, for example proteinuric ones, in order to reduce the severity of these conditions. AS PUFA supplementation could worsen dyslipidemia, a balanced intake could be necessary.

English form should be a little revised. 

---------------------------

ARRIVE Guidelines:
Authors should be encouraged to fill the form in the applicable boxes, for example, statistical methods. 

Author Response

Comments 1: The paper is interesting. This field of research actually deserves our attention also to reduce the impact of nephropathies. Methodology and statistical tools are clearly described and the tests employed are simple. I agree with the limitations described by Authors. Further studies are needed to clarify the role of PUFA supplementation in humans with kidney diseases, for example proteinuric ones, in order to reduce the severity of these conditions.

Response: Thank you very much for reviewing our manuscript and providing invaluable insights and suggestions to improve the presentation of our work. We demonstrated that PUFA deficiency worsens the adaptive response of renal proximal tubules to proteinuria. However, this adaptive response is enhanced by PUFA supplementation at the physiological intake level. This underscores the importance of appropriate PUFA intake for the homeostasis of tubular function. However, this study did not examine the usefulness of PUFA supplementation beyond the physiological intake level; therefore, the appropriate extent of PUFA supplementation in patients with kidney disease with proteinuria remains unelucidated. In the revised manuscript, we have clarified that this study evaluated PUFA supplementation at the physiological intake level. (Page1, line44; Page 8, line 258-259, line 262; Page 12, line 358; Page 14, line 388-399; Page15, line 419; Page 17, line 440).

However, our theory that appropriate PUFA intake is necessary in patients with kidney disease accompanied by proteinuria has not been verified and needs to be further investigated. Especially, the usefulness of PUFA supplementation beyond the physiological intake level was not verified. We have included these details in the discussion section of the revised manuscript. (Page19, line 504- Page 19, line 515).

Comments 2: AS PUFA supplementation could worsen dyslipidemia, a balanced intake could be necessary.

Response: Thank you for pointing this out. We have added an explanation regarding the significance of balanced PUFA intake in the discussion section. As mentioned above, the appropriate extent of PUFA supplementation for patients with kidney disease-induced proteinuria remains undetermined. Excessive n-6 PUFA intake can lead to dyslipidemia, including increased LDL and decreased HDL concentrations, LDL oxidation, and inflammatory stimulation owing to the activation of the arachidonic acid pathway, which may increase the risk of CVD. Excessive n-3 PUFA intake induces bleeding tendencies and excessive immunosuppression. Excessive n-6 PUFA intake is particularly problematic, and a high n-6: n-3 ratio correlates with CVD. In general, appropriate PUFA intake and PUFA balance (n-6: n-3 <4:1) are necessary for maintaining health. As excessive or unbalanced PUFA intake may induce adverse events, appropriate balanced PUFA intake is crucial. However, the appropriate PUFA intake level and PUFA balance for patients with kidney disease remain unelucidated and constitute a topic for further research. (Page 19, line 531-546).

Comments 3: English form should be a little revised.

Response: Per your suggestion, the revised manuscript has been rechecked and proofread by a native English speaker.

Comments 4: Authors should be encouraged to fulfill the form in the appliable boxes, for example statistical methods.

Response: Thank you for your suggestion. We have revised the ARRIVE form accordingly.

Reviewer 2 Report

Comments and Suggestions for Authors

Nice experiment, but its translation to clinical practice is questionable. The authors should address whether PUFA deficiency is a clinical problem in humans. Please describe the epidemiological aspects of PUFA deficiency from the world perspective.

The provided data are not supporting PUFA supplementation. Therefore other then defficience mechanism may explain benefits reporter in studies with supplementation.  This aspect should be better disccussed.

Minor:

Please change the description of Table 1 - Physiological data seems not appropriate. In addition decrease the accurate of data presentation.

Author Response

Comments1: Nice experiment, but its translation to clinical practice is questionable. The authors should address whether PUFA deficiency is a clinical problem in humans. Please describe the epidemiological aspects of PUFA deficiency from the world perspective.

Response: Thank you very much for reviewing our manuscript and providing invaluable insight and suggestions. We concur that PUFA deficiency rarely causes clinical problems in humans who can consume diverse diets, which include vegetable oils and fish that provide adequate PUFAs. However, PUFA deficiencies may occur in individuals who are on extremely low-fat diets, those with anorexia, and those who undertake long-term stringent dietary restrictions, such as in some monasteries and religious institutions. PUFA deficiencies can be induced by digestive diseases that cause fat malabsorption (e.g., inflammatory bowel disease, chronic pancreatitis with exocrine insufficiency, liver cirrhosis, and bile secretion insufficiency) as well as by iatrogenic factors, such as long-term dependence on enteral or central intravenous nutrition with inadequate lipid supply. Furthermore, PUFA deficiency can occur in geographical regions such as sub-Saharan Africa (Ethiopia, Somalia), certain parts of South Asia (Bangladesh, Nepal, Himalayan region of Tibet, Bhutan, Sri Lanka, and the rural areas of India), and war- and conflict-affected regions (Syria, Yemen), which are afflicted by malnutrition and low fat intake owing to chronic extreme food shortages, poverty, and a diet that predominantly comprises carbohydrates. Under these special conditions, the renal tubular functional maladaptation that was elucidated in this study may occur. Kidney disease is a common disease that can occur in individuals in various regions and across diverse dietary environments. Therefore, understanding the effects of PUFA deficiency on renal tubular function is important to elucidate the pathology of different patients with kidney diseases. We have included these details in the discussion section of the revised manuscript. (Page 19, line 548-Page20, line566).

Comments 2: The provided data are not supporting PUFA supplementation. Therefore, other than deficiency mechanism may explain benefits reported in studies with supplementation. This aspect should be better discussed.

Response: Thank you for pointing this out. PUFA supplementation confers favorable clinical effects in various kidney diseases. In this study, PUFA supplementation was performed to revert from a state of PUFA deficiency to that of physiological PUFA levels, and the effect of this intervention on improved renal tubular adaptive functional response against proteinuria was detected. However, the current study did not examine the various favorable clinical effects that have been reported in previous studies examining the effects of PUFA supplementation beyond the physiological intake level in kidney diseases. The clinical effects of PUFA supplementation beyond physiological intake level appear to be mediated by mechanisms other than those associated with improving PUFA deficiency. Therefore, a different experimental design will be necessary to verify these effects in further research. We have included these details in the discussion section of the revised manuscript. (Page 19, line 511-546).

Minor: Please change the description of Table 1 - Physiological data seems not appropriate. In addition decrease the accurate of data presentation.

Response: Thank you for pointing this out. We have changed the description of Table 2 as follows: “Physiological data” to “Changes in body and organ weight.” (Page 7, line 243). We rounded the data to one decimal place to decrease the accurate of data presentation.

Reviewer 3 Report

Comments and Suggestions for Authors

Title: PUFA Supports Renal PPARα Function and Energy Homeostasis and Facilitates the Adaptive Tubular Endocytosis Reabsorption Response Against Urinary Protein Overload

Dear Authors,

First of all, I would like to express my sincere gratitude for the opportunity to contribute my opinion to the evaluation of your manuscript. I found the topic addressed extremely interesting, but there are elements that absolutely need to be improved to facilitate the completion of the peer review process.

Title: I suggest shortening it and being clearer about fundamental elements such as the type of study conducted and providing a clear reference to the study in animal models.

Editing: The proposed tables (particularly Tables 1 and 2) are difficult to read, and the figures could also benefit from graphical improvements (the last one, in full discussion, is strongly discouraged).

Abstract: I would suggest placing more emphasis on the potential clinical implications and the possible effects on humans.

Keywords: OK.

Introduction: I believe it could be improved with a broader epidemiological approach to the phenomenon in question, both globally and locally. The objectives (as I interpret them from lines 80 to 84) are poorly stated and unclear. I suggest using the standard phrasing: "The primary objectives of the study were... while the secondary objectives were..." perhaps specifying them with appropriate research questions. I would also suggest adopting a specific section in the methods, e.g., "Aims and Research Questions."

Methods: I do not find in the text (nor in the bibliography the relevant reference citation) the validated reporting system correctly used by the authors. Additionally, for the validity and reproducibility of the study, registration in a reference database (e.g., https://www.animalstudyregistry.org/asr_web/index.action) is missing, which would certainly allow the authors to provide all study elements to the relevant scientific community.

Results: This is undoubtedly the strength of the study, which does not require significant suggestions, although it would benefit from the previous recommendations.

Discussion: In its current form (see the following suggestion), they are not supported by up-to-date reference scientific evidence. Consider the necessary update.

Bibliography: It should be expanded according to the previous suggestions and perhaps updated for references older than 15/20 years unless they are methodological or have strong evidence of impact. Some of the references go back as far as 30/40 years, and I do not believe all of them are methodological.

In summary, the manuscript presents scientifically interesting results, but in its current form, there are critical elements that need improvement.

Author Response

Comments 1: First of all, I would like to express my sincere gratitude for the opportunity to contribute my opinion to the evaluation of your manuscript. I found the topic addressed extremely interesting, but there are elements that absolutely need to be improved to facilitate the completion of the peer review process.

Response: Thank you very much for reviewing our manuscript and providing invaluable insight and suggestions. Accordingly, we have revised the manuscript based on your comments.

Comments 2: Title: I suggest shortening it and being clearer about fundamental elements such as the type of study conducted and providing a clear reference to the study in animal models.

Response: Thank you for pointing this out. We changed the title “PUFA Supports Renal PPARα Function and Energy Homeostasis and Facilitate the Adaptive Tubular Endocytosis Reabsorption Response Against Urinary Protein Overload” to “Dietary PUFA Deficiency Impairs Renal Lipid Metabolism and Adaptive Response to Proteinuria in Murine Renal Tubules.”

Comments 3: Editing: The proposed tables (particularly Tables 1 and 2) are difficult to read, and the figures could also benefit from graphical improvements (the last one, in full discussion, is strongly discouraged).

Response: Thank you for your feedback. Following your recommendation, we have revised all figures and tables to enhance clarity and readability.

Comments 4: Abstract: I would suggest placing more emphasis on the potential clinical implications and the possible effects on humans.

Response: Thank you for pointing this out. In the abstract of the revised manuscript, we have discussed the potential clinical implications and the possible effects on humans. (Page 1, lines 32-33, 45-47).

Comments 5: Keywords: OK.

Response: Thank you for your evaluation. We have accordingly retained the original keywords in the revised manuscript.

Comments 6: Introduction: I believe it could be improved with a broader epidemiological approach to the phenomenon in question, both globally and locally. The objectives (as I interpret them from lines 80 to 84) are poorly stated and unclear. I suggest using the standard phrasing: "The primary objectives of the study were... while the secondary objectives were..." perhaps specifying them with appropriate research questions. I would also suggest adopting a specific section in the methods, e.g., "Aims and Research Questions."

Response: Thank you for your feedback. We have revised the introduction and materials and methods to clarify the purpose of this study. In the revised materials and methods section, we incorporated a dedicated section titled "Aims and Research Questions." This specific section outlines the primary and secondary objectives in our research. (Page 2, line 78-Page 3, line 100).

Comments 7: Methods: I do not find in the text (nor in the bibliography the relevant reference citation) the validated reporting system correctly used by the authors. Additionally, for the validity and reproducibility of the study, registration in a reference database (e.g.,https://www.animalstudyregistry.org/asr_web/index.action) is missing, which would certainly allow the authors to provide all study elements to the relevant scientific community.

Response: Thank you for pointing this out. We have revised the reference accordingly. However, the present study was not registered in an animal study registry because it was not an essential requirement for study conduct.

Comments 8: Results: This is undoubtedly the strength of the study, which does not require significant suggestions, although it would benefit from the previous recommendations.

Response: Thank you for your evaluation.

Comments 9: Discussion: In its current form (see the following suggestion), they are not supported by up-to-date reference scientific evidence. Consider the necessary update.

Bibliography: It should be expanded according to the previous suggestions and perhaps updated for references older than 15/20 years unless they are methodological or have strong evidence of impact. Some of the references go back as far as 30/40 years, and I do not believe all of them are methodological.

Response: Thank you for your insightful comments. We have updated the references to include more recent references.

Comments 10: In summary, the manuscript presents scientifically interesting results, but in its current form, there are critical elements that need improvement.

Response: Thank you. We have revised the manuscript appropriately based on your comments.

Reviewer 4 Report

Comments and Suggestions for Authors

The study explores the role of PUFA in renal energy balance but does not clearly describe the molecular processes connecting PUFA deficit to autophagy activation and compromised protein reabsorption. A more detailed examination of the mechanistic interplay of these elements would enhance the study.

The manuscript includes various statistical comparisons; nevertheless, it fails to indicate if tests for normality and post-hoc corrections were conducted. 

The study mainly compared PUFA-deficient and PUFA-supplemented groups; however, a more stringent control group (such as an additional group with an isocaloric diet substitution) should be considered to eliminate confounding variables like total fat consumption rather than simply PUFA shortage.

The work indicates that PUFA deficit results in elevated lysosomes and autophagy; however, it does not directly evaluate autophagic flux by the use of inhibitors such as Bafilomycin A1 or LC3 turnover assays. Incorporating these experiments would strengthen the idea that autophagy is activated rather than inhibited. The author need to explain it in the discussion.

Western blot is too small size. 

Author Response

Comments 1: The study explores the role of PUFA in renal energy balance but does not clearly describe the molecular processes connecting PUFA deficit to autophagy activation and compromised protein reabsorption. A more detailed examination of the mechanistic interplay of these elements would enhance the study.

Response: Thank you for your feedback. Based on the results of our previous study, we have further clarified the molecular mechanism of PUFA deficiency and autophagy activation. An explanation of the interrelationships of renal energy depletion, endocytosis–lysosome flux, and increased urinary protein excretion has been included in the Results section of the revised manuscript. (Page 12, line 328-346).

Comments 2: The manuscript includes various statistical comparisons; nevertheless, it fails to indicate if tests for normality and post-hoc corrections were conducted. 

Response: Thank you for pointing this out. We have described the normality assessment in the methods section, and the original manuscript comprised a description of intergroup differences that were ascertained using ANOVA, followed by Tukey's post-hoc test. (Page 7, line 230-231).

Comments 3: The study mainly compared PUFA-deficient and PUFA-supplemented groups; however, a more stringent control group (such as an additional group with an isocaloric diet substitution) should be considered to eliminate confounding variables like total fat consumption rather than simply PUFA shortage.

Response: Thank you for your comment. To ensure the same calorie intake and total fat consumption in the PUFA-deficient and PUFA-supplemented groups, mice in each group received the same calories from PUFA-deficient or PUFA-containing oil, in addition to the same PUFA-deficient diet. (Page 2, line 93-Page3, line100). Therefore, we believed that the study design adequately accounts for potential confounding effects related to calorie intake and total fat consumption.

Comments 4: The work indicates that PUFA deficit results in elevated lysosomes and autophagy; however, it does not directly evaluate autophagic flux by the use of inhibitors such as Bafilomycin A1 or LC3 turnover assays. Incorporating these experiments would strengthen the idea that autophagy is activated rather than inhibited. The author need to explain it in the discussion.

Response: Thank you for pointing this out. Although we recognize the need for these experiments, this confirmation is not feasible within the specified revision period. Therefore, we have mentioned this aspect as a limitation that warrants further investigation. (Page 20, line 581-583).

Comments 5: Western blot is too small size. 

Response: Thank you for pointing this out. We have modified the figure to enhance the size of the western blot-derived images.

Round 2

Reviewer 2 Report

Comments and Suggestions for Authors

The paper was proved.

I am surprised that the authors put the study aims in the Materials and Methods section. They should be presented as the last part of the introduction with the support of why they were formulated.

Author Response

Comment: The paper was proved. I am surprised that the authors put the study aims in the Materials and Methods section. They should be presented as the last part of the introduction with the support of why they were formulated.

Response: Thank you very much for reviewing our manuscript and providing invaluable insight and suggestions. We have moved the description of study aims from the material section to the introduction section.

Reviewer 3 Report

Comments and Suggestions for Authors

Dear Authors,

in this form certly almost ready for publication but rest conflict of reporting tool usign (Arrive) in the text (methods) and in the references.

Best

Author Response

Comment: Dear Authors, in this form curtly almost ready for publication but rest conflict of reporting tool usign (Arrive) in the text (methods) and in the references.

Response: Thank you for your suggestion. We have added the insufficient information in the revised manuscript, and have revised the ARRIVE form accordingly.